# Learning visual biases from human imagination

**Carl Vondrick**   **Hamed Pirsiavash**†   **Aude Oliva**   **Antonio Torralba**
Massachusetts Institute of Technology     †University of Maryland, Baltimore County
{vondrick,oliva,torralba}@mit.edu   hpirsiav@umbc.edu

## Abstract

Although the human visual system can recognize many concepts under challenging conditions, it still has some biases. In this paper, we investigate whether we can extract these biases and transfer them into a machine recognition system. We introduce a novel method that, inspired by well-known tools in human psychophysics, estimates the biases that the human visual system might use for recognition, but in computer vision feature spaces. Our experiments are surprising, and suggest that classifiers from the human visual system can be transferred into a machine with some success. Since these classifiers seem to capture favorable biases in the human visual system, we further present an SVM formulation that constrains the orientation of the SVM hyperplane to agree with the bias from human visual system. Our results suggest that transferring this human bias into machines may help object recognition systems generalize across datasets and perform better when very little training data is available.

## 1   Introduction

Computer vision researchers often go through great lengths to remove dataset biases from their models [32, 20]. However, not all biases are adversarial. Even natural recognition systems, such as the human visual system, have biases. Some of the most well known human biases, for example, are the canonical perspective (prefer to see objects from a certain perspective) [26] and Gestalt laws of grouping (tendency to see objects in collections of parts) [11].

We hypothesize that biases in the human visual system can be beneficial for visual understanding. Since recognition is an underconstrained problem, the biases that the human visual system developed may provide useful priors for perception. In this paper, we develop a novel method to learn some biases from the human visual system and incorporate them into computer vision systems.

We focus our approach on learning the biases that people may have for the appearance of objects. To illustrate our method, consider what may seem like an odd experiment. Suppose we sample i.i.d. white noise from a standard normal distribution, and treat it as a point in a visual feature space, e.g. CNN or HOG. What is the chance that this sample corresponds to visual features of a car image? Fig.1a visualizes some samples [35] and, as expected, we see noise. But, let us not stop there. We next generate one hundred fifty thousand points from the same distribution, and ask workers on Amazon Mechanical Turk to classify visualizations of each sample as a car or not. Fig.1c visualizes the average of visual features that workers believed were cars. Although our dataset consists of only white noise, a car emerges!

Sampling noise may seem unusual to computer vision researchers, but a similar procedure, named classification images, has gained popularity in human psychophysics [2] for estimating an approximate template the human visual system internally uses for recognition [18, 4]. In the procedure, an observer looks at an image perturbed with random noise and indicates whether they perceive a target category. After a large number of trials, psychophysics researchers can apply basic statistics to extract an approximation of the internal template the observer used for recognition. Since the

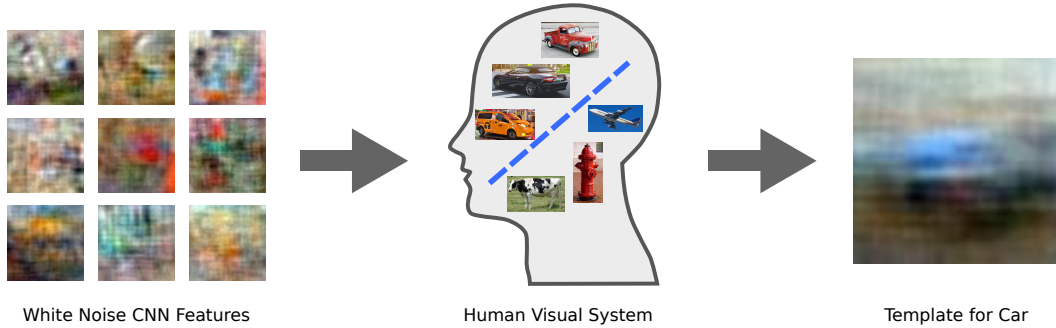

White Noise CNN Features          Human Visual System          Template for Car

Figure 1: Although all image patches on the left are just noise, when we show thousands of them to online workers and ask them to find ones that look like cars, a car emerges in the average, shown on the right. This noise-driven method is based on well known tools in human psychophysics that estimates the biases that the human visual system uses for recognition. We explore how to transfer these biases into a machine.

procedure is done with noise, the estimated template reveals some of the cues that the human visual system used for discrimination.

We propose to extend classification images to estimate biases from the human visual system. However, our approach makes two modifications. Firstly, we estimate the template in state-of-the-art computer vision feature spaces [8, 19], which allows us to incorporate these biases into learning algorithms in computer vision systems. To do this, we take advantage of algorithms that invert visual features back to images [35]. By estimating these biases in a feature space, we can learn biases for how humans may correspond mid-level features, such as shapes and colors, with objects. To our knowledge, we are the first to estimate classification images in vision feature spaces. Secondly, we want our template to be biased by the human visual system and not our choice of dataset. Unlike classification images, we do not perturb real images; instead our approach only uses visualizations of feature space noise to estimate the templates. We capitalize on the ability of people to discern visual objects from random noise in a systematic manner [16].

## 2 Related Work

**Mental Images**: Our methods build upon work to extract mental images from a user's head for both general objects [15], faces [23], and scenes [17]. However, our work differs because we estimate mental images in state-of-the-art computer vision feature spaces, which allows us to integrate the mental images into a machine recognition system.

**Visual Biases:** Our paper studies biases in the human visual system similar to [26, 11], but we wish to transfer these biases into a computer recognition system. We extend ideas [24] to use computer vision to analyze these biases. Our work is also closely related to dataset biases [32, 28], which motivates us to try to transfer favorable biases into recognition systems.

**Human-in-the-Loop:** The idea to transfer biases from the human mind into object recognition is inspired by many recent works that puts a human in the computer vision loop [6, 27], trains recognition systems with active learning [33], and studies crowdsourcing [34, 31]. The primary difference of these approaches and our work is, rather than using crowds as a workforce, we want to extract biases from the worker's visual systems.

**Feature Visualization:** Our work explores a novel application of feature visualizations [36, 35, 22]. Rather than using feature visualizations to diagnose computer vision systems, we use them to inspect and learn biases in the human visual system.

**Transfer Learning:** We also build upon methods in transfer learning to incorporate priors into learning algorithms. A common transfer learning method for SVMs is to change the regularization term $||w||_2^2$ to $||w - c||_2^2$ where $c$ is the prior [29, 37]. However, this imposes a prior on both the norm and orientation of $w$. In our case, since the visual bias does not provide an additional prior on the norm, we present a SVM formulation that constrains only the orientation of $w$ to be close to $c$.

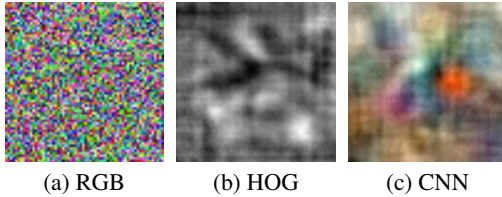

(a) RGB      (b) HOG      (c) CNN

Figure 2: We visualize white noise in RGB and feature spaces. To visualize white noise features, we use feature inversion algorithms [35]. White noise in feature space has correlations in image space that white noise in RGB does not. We capitalize on this structure to estimate visual biases in feature space without using real images.

Our approach extends sign constraints on SVMs [12], but instead enforces orientation constraints. Our method enforces a hard orientation constraint, which builds on soft orientation constraints [3].

## 3 Classification Images Review

The procedure *classification images* is a popular method in human psychophysics that attempts to estimate the internal template that the human visual system might use for recognition of a category [18, 4]. We review classification images in this section as it is the inspiration for our method.

The goal is to approximate the template $\tilde{c} \in \mathbb{R}^d$ that a human observer uses to discriminate between two classes $A$ and $B$, e.g. male vs. female faces, or chair vs. not chair. Suppose we have intensity images $a \in A \subseteq \mathbb{R}^d$ and $b \in B \subseteq \mathbb{R}^d$. If we sample white noise $\epsilon \sim \mathcal{N}(0^d, I_d)$ and ask an observer to indicate the class label for $a + \epsilon$, most of the time the observer will answer with the correct class label $A$. However, there is a chance that $\epsilon$ might manipulate $a$ to cause the observer to mistakenly label $a + \epsilon$ as class $B$.

The insight into classification images is that, if we perform a large number of trials, then we can estimate a decision function $f(\cdot)$ that discriminates between $A$ and $B$, but makes the same mistakes as the observer. Since $f(\cdot)$ makes the same errors, it provides an estimate of the template that the observer internally used to discriminate $A$ from $B$. By analyzing this model, we can then gain insight into how a visual system might recognize different categories.

Since psychophysics researchers are interested in models that are interpretable, classification images are often linear approximations of the form $f(x; \tilde{c}) = \tilde{c}^T x$. The template $\tilde{c} \in \mathbb{R}^d$ can be estimated in many ways, but the most common is a sum of the stimulus images:

$$\tilde{c} = (\mu_{AA} + \mu_{BA}) - (\mu_{AB} + \mu_{BB}) \tag{1}$$

where $\mu_{XY}$ is the average image where the true class is $X$ and the observer predicted class $Y$. The template $c$ is fairly intuitive: it will have large positive value on locations that the observer used to predict $A$, and large negative value for locations correlated with predicting $B$. Although classification images is simple, this procedure has led to insights in human perception. For example, [30] used classification images to study face processing strategies in the human visual system. For a complete analysis of classification images, we refer readers to review articles [25, 10].

## 4 Estimating Human Biases in Feature Spaces

Standard classification images is performed with perturbing real images with white noise. However, this approach may negatively bias the template by the choice of dataset. Instead, we are interested in estimating templates that capture biases in the human visual system and not datasets.

We propose to estimate these templates by only sampling white noise (with no real images). Unfortunately, sampling just white noise in RGB is extremely unlikely to result in a natural image (see Fig.2a). To overcome this, we can estimate the templates in feature spaces [8, 19] used in computer vision. Feature spaces encode higher abstractions of images (such as gradients, shapes, or colors). While sampling white noise in feature space may still not lay on the manifold of natural images, it is more likely to capture statistics relevant for recognition. Since humans cannot directly interpret abstract feature spaces, we can use feature inversion algorithms [35, 36] to visualize them.

Using these ideas, we first sample noise from a zero-mean, unit-covariance Gaussian distribution $x \sim \mathcal{N}(0_d, I_d)$. We then invert the noise feature $x$ back to an image $\phi^{-1}(x)$ where $\phi^{-1}(\cdot)$ is the

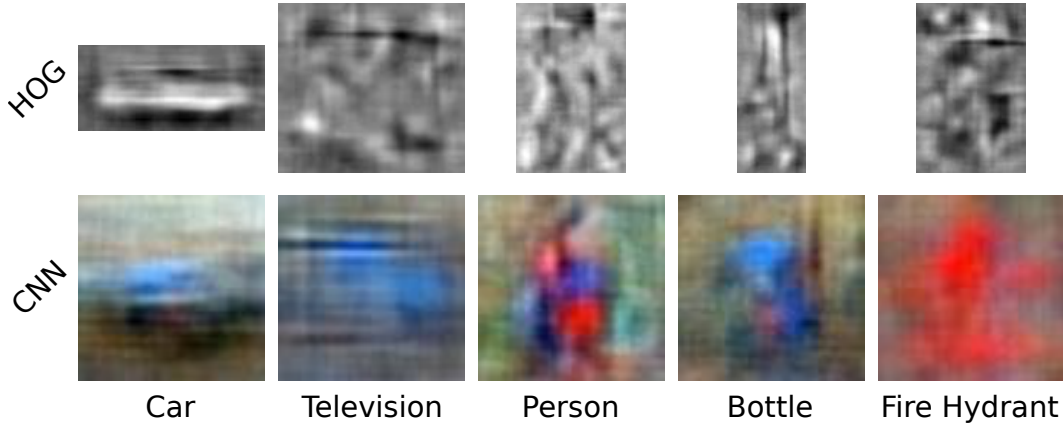

Figure 3: We visualize some biases estimated from trials by Mechanical Turk workers.

feature inverse. By instructing people to indicate whether a visualization of noise is a target category or not, we can build a linear template $c \in \mathbb{R}^d$ that approximates people's internal templates:

$$c = \mu_A - \mu_B \tag{2}$$

where $\mu_A \in \mathbb{R}^d$ is the average, in feature space, of white noise that workers incorrectly believe is the target object, and similarly $\mu_B \in \mathbb{R}^d$ is the average of noise that workers believe is noise.

Eqn.2 is a special case of the original classification images Eqn.1 where the background class $B$ is white noise and the positive class $A$ is empty. Instead, we rely on humans to hallucinate objects in noise to form $\mu_A$. Since we build these biases with only white Gaussian noise and no real images, our approach may be robust to many issues in dataset bias [32]. Instead, templates from our method can inherit the biases for the appearances of objects present in the human visual system, which we suspect provides advantageous signals about the visual world.

In order to estimate $c$ from noise, we need to perform many trials, which we can conduct effectively on Amazon Mechanical Turk [31]. We sampled $150,000$ points from a standard normal multivariate distribution, and inverted each sample with the feature inversion algorithm from HOGgles [35]. We then instructed workers to indicate whether they see the target category or not in the visualization. Since we found that the interpretation of noise visualizations depends on the scale, we show the worker three different scales. We paid workers 10¢ to label 100 images, and workers often collectively solved the entire batch in a few hours. In order to assure quality, we occasionally gave workers an easy example to which we knew the answer, and only retained work from workers who performed well above chance. We only used the easy examples to qualify workers, and discarded them when computing the final template.

## 5 Visualizing Biases

Although subjects are classifying zero-mean, identity covariance white Gaussian noise with no real images, objects can emerge after many trials. To show this, we performed experiments with both HOG [8] and the last convolutional layer (`pool5`) of a convolutional neural network (CNN) trained on ImageNet [19, 9] for several common object categories. We visualize some of the templates from our method in Fig.3. Although the templates are blurred, they seem to show significant detail about the object. For example, in the car template, we can clearly see a vehicle-like object in the center sitting on top of a dark road and lighter sky. The television template resembles a rectangular structure, and the fire hydrant templates reveals a red hydrant with two arms on the side. The templates seem to contain the canonical perspective of objects [26], but also extends them with color and shape biases.

In these visualizations, we have assumed that all workers on Mechanical Turk share the same appearance bias of objects. However, this assumption is not necessarily true. To examine this, we instructed workers on Mechanical Turk to find "sport balls" in CNN noise, and clustered workers by their geographic location. Fig.4 shows the templates for both India and the United States. Even

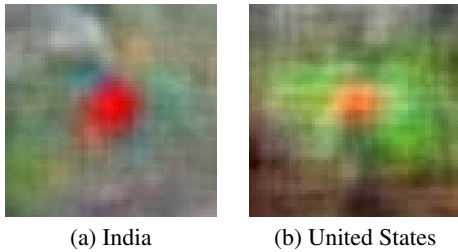

(a) India       (b) United States

Figure 4: We grouped users by their geographic location (US or India) and instructed each group to classify CNN noise as a sports ball or not, which allows us to see how biases can vary by culture. Indians seem to imagine a red ball, which is the standard color for a cricket ball and the predominant sport in India. Americans seem to imagine a brown or orange ball, which could be an American football or basketball, both popular sports in the U.S.

though both sets of workers were labeling noise from the same distribution, Indian workers seemed to imagine red balls, while American workers tended to imagine orange/brown balls. Remarkably, the most popular sport in India is cricket, which is played with a red ball, and popular sports in the United States are American football and basketball, which are played with brown/orange balls. We conjecture that Americans and Indians may have different mental images of sports balls in their head and the color is influenced by popular sports in their country. This effect is likely attributed to phenomena in social psychology where human perception can be influenced by culture [7, 5]. Since environment plays a role in the development of the human vision system, people from different cultures likely develop slightly different images inside their head.

## 6 Leveraging Humans Biases for Recognition

If the biases we learn are beneficial for recognition, then we would expect them to perform above chance at recognizing objects in real images. To evaluate this, we use the visual biases $c$ directly as a classifier for object recognition. We quantify their performance on object classification in real-world images using the PASCAL VOC 2011 dataset [13], evaluating against the validation set. Since PASCAL VOC does not have a fire hydrant category, we downloaded 63 images from Flickr with fire hydrants and added them to the validation set. We report performance as the average precision on a precision-recall curve.

The results in Fig.5 suggest that biases from the human visual system do capture some signals useful for classifying objects in real images. Although the classifiers are estimated using only white noise, in most cases the templates are significantly outperforming chance, suggesting that biases from the human visual system may be beneficial computationally.

Our results suggest that shape is an important bias to discriminate objects in CNN feature space. Notice how the top classifications in Fig.6 tend to share the same rough shape by category. For example, the classifier for person finds people that are upright, and the television classifier fires on rectangular shapes. The confusions are quantified Fig.7: bottles are often confused as people, and cars are confused as buses. Moreover, some templates appear to rely on color as well. Fig.6 suggests that the classifier for fire-hydrant correctly favors red objects, which is evidenced by it frequently firing on people wearing red clothes. The bottle classifier seems to be incorrectly biased towards blue objects, which contributes to its poor performance.

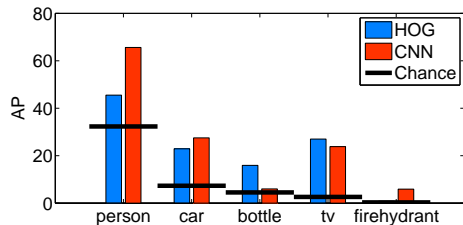

|  | car | person | f-hydrant | bottle | tv |
|---|---|---|---|---|---|
| HOG | 22.9 | 45.5 | 0.8 | 15.9 | 27.0 |
| CNN | 27.5 | 65.6 | 5.9 | 6.0 | 23.8 |
| Chance | 7.3 | 32.3 | 0.3 | 4.5 | 2.6 |

Figure 5: We show the average precision (AP) for object classification on PASCAL VOC 2011 using templates estimated with noise. Even though the template is created without a dataset, it performs significantly above chance.

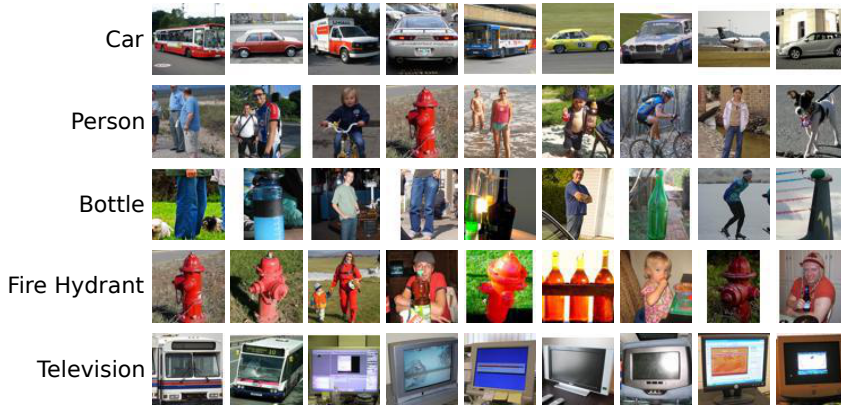

Figure 6: We show some of the top classifications from the human biases estimated with CNN features. Note that real data is not used in building these models.

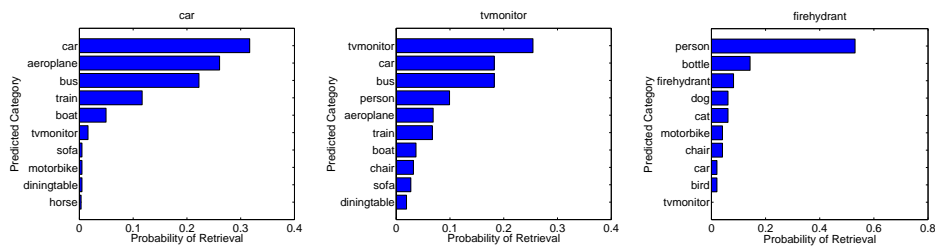

Figure 7: We plot the class confusions for some human biases on top classifications with CNN features. We show only the top 10 classes for visualization. Notice that many of the confusions may be sensible, e.g. the classifier for car tends to retrieve vehicles, and the fire hydrant classifier commonly mistakes people and bottles.

While the motivation of this experiment has been to study whether human biases are favorable for recognition, our approach has some applications. Although templates estimated from white noise will likely never be a substitute for massive labeled datasets, our approach can be helpful for recognizing objects when no training data is available. Rather, our approach enables us to build classifiers for categories that a person has only imagined and never seen. In our experiments, we evaluated on common categories to make evaluation simpler, but in principle our approach can work for rare categories as well. We also wish to note that the CNN features used here are trained to classify images on ImageNet [9] LSVRC 2012, and hence had access to data. However, we showed competitive results for HOG as well, which is a hand-crafted feature, as well as results for a category that the CNN network did not see during training (fire hydrants).

## 7 Learning with Human Biases

Our experiments to visualize the templates and use them as object recognition systems suggest that visual biases from the human visual system provide some signals that are useful for discriminating objects in real world images. In this section, we investigate how to incorporate these signals into learning algorithms when there is some training data available. We present an SVM that constrains the separating hyperplane to have an orientation similar to the human bias we estimated.

### 7.1 SVM with Orientation Constraints

Let $x_i \in \mathbb{R}^m$ be a training point and $y_i \in \{-1, 1\}$ be its label for $1 \leq i \leq n$. A standard SVM seeks a separating hyperplane $w \in \mathbb{R}^m$ with a bias $b \in \mathbb{R}$ that maximizes the margin between positive and negative examples. We wish to add the constraint that the SVM hyperplane $w$ must be at most $\cos^{-1}(\theta)$ degrees away from the bias template $c$:

$$\min_{w,b,\xi} \frac{\lambda}{2} w^T w + \sum_{i=1}^{n} \xi_i \quad \text{s.t.} \quad y_i\left(w^T x_i + b\right) \geq 1 - \xi_i, \ \xi_i \geq 0 \quad \text{(3a)}$$

$$\theta \leq \frac{w^T c}{\sqrt{w^T w}} \quad \text{(3b)}$$

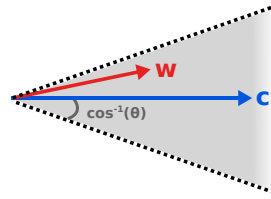

Figure 8

where $\xi_i \in \mathbb{R}$ are the slack variables, $\lambda$ is the regularization hyper-parameter, and Eqn.3b is the orientation prior such that $\theta \in (0, 1]$ bounds the maximum angle that the $w$ is allowed to deviate from $c$. Note that we have assumed, without loss of generality, that $||c||_2 = 1$. Fig.8 shows a visualization of this orientation constraint. The feasible space for the solution is the grayed hypercone. The SVM solution $w$ is not allowed to deviate from the prior classifier $c$ by more than $\cos^{-1}(\theta)$ degrees.

## 7.2 Optimization

We optimize Eqn.3 efficiently by writing the objective as a conic program. We rewrite Eqn.3b as $\sqrt{w^T w} \leq \frac{w^T c}{\theta}$ and introduce an auxiliary variable $\alpha \in \mathbb{R}$ such that $\sqrt{w^T w} \leq \alpha \leq \frac{w^T c}{\theta}$. Substituting these constraints into Eqn.3 and replacing the SVM regularization term with $\frac{\lambda}{2}\alpha^2$ leads to the conic program:

$$\min_{w,b,\xi,\alpha} \frac{\lambda}{2}\alpha^2 + \sum_{i=1}^{n} \xi_i \quad \text{s.t.} \quad y_i\left(w^T x_i + b\right) \geq 1 - \xi_i, \quad \xi_i \geq 0, \quad \sqrt{w^T w} \leq \alpha \quad \text{(4a)}$$

$$\alpha \leq \frac{w^T c}{\theta} \quad \text{(4b)}$$

Since at the minimum $a^2 = w^T w$, Eqn.4 is equivalent to Eqn.3, but in a standard conic program form. As conic programs are convex by construction, we can then optimize it efficiently using off-the-shelf solvers, which we use MOSEK [1]. Note that removing Eqn.4b makes it equivalent to the standard SVM. $\cos^{-1}(\theta)$ specifies the angle of the cone. In our experiments, we found $30°$ to be reasonable. While this angle is not very restrictive in low dimensions, it becomes much more restrictive as the number of dimensions increases [21].

## 7.3 Experiments

We previously used the bias template as a classifier for recognizing objects when there is no training data available. However, in some cases, there may be a few real examples available for learning. We can incorporate the bias template into learning using an SVM with orientation constraints. Using the same evaluation procedure as the previous section, we compare three approaches: 1) a single SVM trained with only a few positives and the entire negative set, 2) the same SVM with orientation priors for $\cos(\theta) = 30°$ on the human bias, and 3) the human bias alone. We then follow the same experimental setup as before. We show full results for the SVM with orientation priors in Fig.9. In general, biases from the human visual system can assist the SVM when the amount of positive training data is only a few examples. In these low data regimes, acquiring classifiers from the human visual system can improve performance with a margin, sometimes 10% AP.

Furthermore, standard computer vision datasets often suffer from dataset biases that harm cross dataset generalization performance [32, 28]. Since the template we estimate is biased by the human visual system and not datasets (there is no dataset), we believe our approach may help cross dataset generalization. We trained an SVM classifier with CNN features to recognize cars on Caltech 101 [14], but we tested it on object classification with PASCAL VOC 2011. Fig.10a suggest that, by constraining the SVM to be close to the human bias for car, we are able to improve the generalization performance of our classifiers, sometimes over 5% AP. We then tried the reverse experiment in Fig.10b: we trained on PASCAL VOC 2011, but tested on Caltech 101. While PASCAL VOC provides a much better sample of the visual world, the orientation priors still help generalization performance when there is little training data available. These results suggest that incorporating the biases from the human visual system may help alleviate some dataset bias issues in computer vision.

|          | 0 positives |       | 1 positive |           | 5 positives |           |
|---------:|:-----------:|:-----:|:----------:|:---------:|:-----------:|:---------:|
| Category | Chance | Human | SVM | SVM+Human | SVM | SVM+Human |
| car | 7.3 | **27.5** | 11.6 | **29.0** | 37.8 | **43.5** |
| person | 32.3 | **65.6** | 55.2 | **69.3** | 70.1 | **73.7** |
| f-hydrant | 0.3 | **5.9** | 1.7 | **7.0** | **50.1** | **50.1** |
| bottle | 4.5 | **6.0** | 11.2 | **11.7** | 38.1 | **38.7** |
| tv | 2.6 | **23.8** | 38.6 | **43.1** | 66.7 | **68.8** |

Figure 9: We show AP for the SVM with orientation priors for object classification on PASCAL VOC 2011 for varying amount of positive data with CNN features. All results are means over random subsamples of the training sets. SVM+Hum refers to SVM with the human bias as an orientation prior.

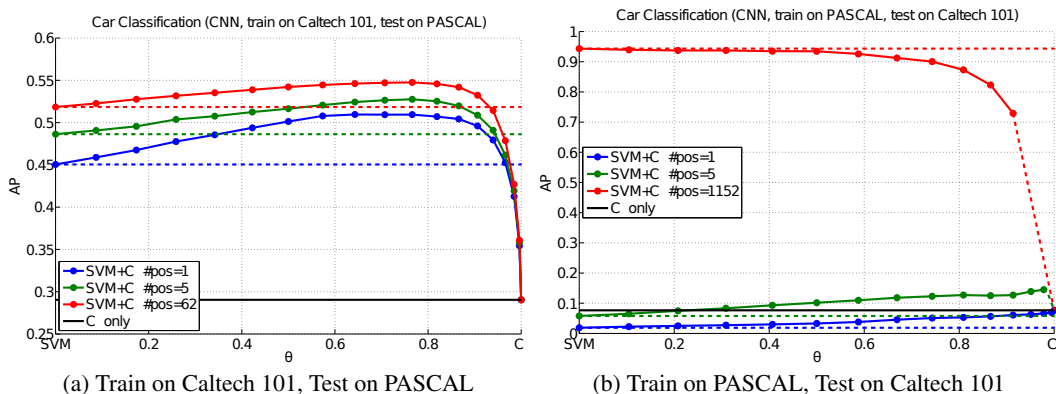

(a) Train on Caltech 101, Test on PASCAL  (b) Train on PASCAL, Test on Caltech 101

Figure 10: Since bias from humans is estimated with only noise, it tends to be biased towards the human visual system instead of datasets. (a) We train an SVM to classify cars on Caltech 101 that is constrained towards the bias template, and evaluate it on PASCAL VOC 2011. For every training set size, constraining the SVM to the human bias with $\theta \approx 0.75$ is able to improve generalization performance. (b) We train a constrained SVM on PASCAL VOC 2011 and test on Caltech 101. For low data regimes, the human bias may help boost performance.

## 8  Conclusion

Since the human visual system is one of the best recognition systems, we hypothesize that its biases may be useful for visual understanding. In this paper, we presented a novel method to estimate some biases that people have for the appearance of objects. By estimating these biases in state-of-the-art computer vision feature spaces, we can transfer these templates into a machine, and leverage them computationally. Our experiments suggest biases from the human visual system may provide useful signals for computer vision systems, especially when little, if any, training data is available.

**Acknowledgements:** We thank Aditya Khosla for important discussions, and Andrew Owens and Zoya Bylinskii for helpful comments. Funding for this research was partially supported by a Google PhD Fellowship to CV, and a Google research award and ONR MURI N000141010933 to AT.

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
