[Reviews · NeurIPS 2015]

Submitted by Assigned_Reviewer_1

This paper present an approach to estimate a template for an object from a set of noise images that were perceived by human as an object. Key aspect of this paper is that noise images are created in feature space and then transformed to images.

The experimental results demonstrate that such template by itself can be used in image classification task. In addition, in regular SVM learning if little training data is available incorporating such template into learning boosts SVM performance. Experimental results are presented on 5 object classes from Caltech 101 and PASCAL VOC 2011 datasets.

Quality: The paper seems technically sound and makes for a nice study of learning visual bias. It would be interesting to see how many noise images per category were actually "recognized" by the instructed workers.

Clarity: The paper is well written and presentation is coherent.

Originality: The paper is incrementally novel.

Significance: The paper covers an interesting topic since it provide intuition on human perception. However, the connection to learning with little examples seems forced. Amount of work required to classify noise images is far more than is required to collect data for regular SVM classifier.
Summary: The paper presents an interesting twist to classification images technique and bridges it to computer vision task of image classification. This connection is indeed impressive, but application (as presented) seems limited due to the huge amount of noise images that needs to be classified.

Submitted by Assigned_Reviewer_2

SUMMARY: The paper directly learns human biases for object classification and uses these biases to improve machine recognition accuracy. The insight is from psychophysics literature which indicates that it is possible to extract (rough) visual models of objects by asking humans to classify random noise images into those that contain an object class and those that don't. In this paper, the authors generate random feature vectors in the HOG or CNN feature space, visualize them using HOGgles, and ask humans to classify the images. The results can be used to obtain a hyperplane separating the target class from others. This hyperplane is used for classification directly as well as to constrain classifier learning.

QUALITY: Overall, the paper is of very high quality: both the experiments and the presentation are very well-thought-out. I had multiple comments and questions as I was reading the paper, but they were all addressed by the authors-- e.g., I had concerns about lack of psychophysics citations but they appeared later in the text; I was going to mention that CNN is pre-trained (the authors kept saying that the results are obtained "without any training data") but this was also discussed in ln. 305-307. Both the machine learning results and the human experiments are solid.

CLARITY: I had no trouble with any part of the paper -- one of the best-written papers I've seen in a while

ORIGINALITY: I've heard of this idea from psychophysics but have not seen this utilized in any way in the machine learning community. Definitely novel work.

SIGNIFICANCE: I think this "out-of-the-box" thinking is very healthy and very necessary for our community.
Summary: This is a great idea, very novel work and solid execution. I greatly enjoyed reading this paper and I think many people in the community will as well.

Submitted by Assigned_Reviewer_3

Summary: This paper draws inspiration from work on psychophysics on classification images.

Large-scale human experiments were run, where people were asked to classify images generated from random noise (randomly generated by inverting HOG or CNN feature spaces to more closely approximate the distribution of natural images).

The results were used to 1) visualize human perception of different classes, 2) see how well classifiers trained on datasets of random noise would work on real images, and 3) use the results as an additional source of information to regularize classifiers trained on a small number of images.

Quality:

This is a very unusual paper.

It is overall a high quality and well written paper where interesting and novel experiments were carried out; however it is unclear if the results or methods of the paper are of practical value.

Clarity:

The paper is well written and clear.

Originality:

This is a really creative paper with an interesting and novel idea.

It is nice that the authors drew in research from other disciplines (psychophysics) that most people in machine learning are unfamiliar with.

Significance:

I think the main value of this paper is that it will inspire readers to think about philosophical issues like what is dataset bias?, does the definition of a class come from human imagination or is inherent to the world?, how does one classify images that lie outside the distribution of natural images?

I think in general people will find it fun and interesting to see how people perceive random images, as well as some of the qualitative results that people from India had different perceptions than people from the US.

At the same time, the claims of the paper are controversial (see points 1 & 2 below) and the methods realistically have little practical value (see point 3 below).

1) Does the proposed method alleviate problems of dataset bias? I don't think so.

I think dataset bias emerges if one trains from datasets/images that come from a different distribution than natural data or problems we ultimately would like to solve (admittedly this is a fuzzy and probably misguided definition). Training a classifier (or generating a classification image) by having people label random white noise in pixel space is training from an extremely different (and also biased) synthetic distribution.

Training from random noise in HOGgles space is similarly another type of biased synthetic dataset.

I think it is an interesting finding to see how people classify images outside the domain of natural images, but this is an entirely different issue than dataset bias.

A machine learning algorithm has biases because of the data that was used to train it as well as limitations of the particular learning model.

Humans have biases due to their personal experiences as well as their learning models (ie., architecture of the brain).

An algorithm trained from human labels will inherently (attempt to) learn to capture the biases of its human labelers.

This isn't altered by using random images as opposed to natural images.

Rather, using random images exacerbates the problem by using an even more unnatural distribution of data.

2) Does the notion of classification images from psychophysics literature offer some new insight into human perception or dataset bias? I don't think so, or at least the paper doesn't go far enough in explaining this concept to communicate this to a NIPS audience that is unfamiliar with psychophysics research (such as myself).

The 2 technical differences in the proposed classification image-based approach compared to methods used in machine learning are a) having people label randomly generated training data instead of natural images, and b) the method of generating classification images to visualize human classifiers.

Looking at cited literature [23,9] classification image literature are making a lot of assumptions about human perception (e.g., a linear separator in feature space) and these assumptions are similar to some models used in machine learning (i.e., similar to a simplified version of Fisher's linear discriminant).

I am unable to grasp why using a random noise training set is desirable or necessary, or if this is just a method of choice to acquire a dataset.

3) The proposed method of regularizing classifiers trained from a small number of images with classification images is creative but impractical.

It required tons of human annotation labelling random images for very small improvements in performance.
Summary: This is an interesting, fun, and creative paper that will inspire readers to think about philosophical issues on dataset bias and human perception.

At the same time, it has some controversial claims and limited practical value.

Submitted by Assigned_Reviewer_4

The method is essentially a twist on a popular psychophyscis method (albeit an interesting one). This is a cool idea but the treatment remains unconvincing. The estimated biases seem obvious (Fig 3): A car is an horizontalish object and pedestrians are verticalish and fire hydrants reddish. Does this really tell us anything about human vision? The application to computer vision is hardly more convincing: Yes -- a reddish template will tend to find most fire hydrants and other reddish things (Fig 6). The classification accuracy is slightly better than chance and I understand that the method can be used to learn categories without training data but it is very hard to imagine (pun intended) that this method will work for object categories for which training examples are not available say on ImageNet.
Summary: This paper describes a method to estimate human biases from observers' imagination. Results are presented for an amazon MT experiment whereby object templates for 5 object categories are estimated. It is shown that the incorporation of these biases in a classification function may improve the accuracy of the system.

Author Feedback
Author rebuttal: Thank you for reading the paper and for your careful comments!

We are glad that the reviewers appreciate the novelty and creativity of the work. Five out of six of the reviewers are generally in favor of the paper. We believe this work can be of wide interest at NIPS because it connects classical ideas in human psychophysics with state-of-the-art visual representations in computer vision.

R4, R6, R8: Computer vision applications. We believe the paper's major contributions are more fundamental by developing new relationships between biases in human vision with computer vision. However, the work also explores two practical computer vision applications. The first application is creating visual classifiers when no training data is available (Sec 6) or only very little (Sec 7.3). We showed quantitative results for recognizing objects without any real training data, as well as for an object category that is missing from both WordNet and ImageNet (fire hydrant). The second application is to help classifiers be robust against some dataset biases (Sec 7.4). Both of these applications have several real-world uses.

R4, R6: Human vision applications. The paper contributes a new method to visualize the biases people have for the shapes/colors/appearances of objects. These human biases are complementary to other biases (such as dataset biases or canonical perspective), which we believe is interesting on its own right. Our approach finds biases that are not obvious, such as the cultural biases for sports balls (Fig 4).

R6: Reason for random noise. We use noise because we do not want to prime annotators.
The biases from the human visual system are subtle. Labeling real images primes the user, which adversely manipulates the acquired images and makes it difficult to study the human biases. For example, if we showed real images of sports balls to Indians/Americans in Fig.4, both populations would recognize sports balls from different cultures, masking human biases. By using noise, we require annotators to use their priors instead, which is able to reveal biases.

R6: Dataset biases. We do not claim to solve all problems of dataset bias. Since our method builds a classifier without using any real-images, we argue our method is robust to many biases arising in real-image datasets (such as how data is collected or downloaded). Instead, our method inherits biases with how people imagine objects from noise. While one could view our approach as using an annotated dataset of noise and therefore has a dataset bias, this is a very particular type of bias that comes from the human visual system and is not obscured by real-image biases.

R1, R6: SVM. Our method requires many human trials, however this is just the cost of creating a classifier that captures human biases. Although this is not the main focus of the paper, we note that our approach for regularizing SVMs with orientation constraints has other applications in transfer learning as well.

R1: Annotation statistics. The number of times a person "recognized" a noise image varied significantly by the worker and the object. However, on average, people indicated they saw an object roughly 20% of the time.

R2, R6: Different classifiers. Our method can be extended to work with more complex classifiers, however this is out-of-scope for this paper. Since this is one of the first papers in computer vision using classification images, we wanted an interpretable model, and linear classifiers are easier to interpret. R6: Nonlinear classification images exist as well, however linear classification images are more popular since they are more interpretable.

R6: Philosophical discussions. We agree that these are very interesting questions! One could view this paper as exploring an alternative definition for objects (how people imagine them), which we believe is an interesting perspective. We hope this paper can spark these discussions at the conference.

R7: Interdisciplinary work. Thank you for your positive comments. Classification images is a classical procedure in psychophysics, and we agree that it is under utilized in machine learning and computer vision. Hence, we believe this paper will interest researchers from several areas at NIPS.

Thank you for consideration.